# pKa Determination of a Histidine Residue in a Short Peptide Using Raman Spectroscopy

**DOI:** 10.3390/molecules24030405

**Published:** 2019-01-23

**Authors:** Brett H. Pogostin, Anders Malmendal, Casey H. Londergan, Karin S. Åkerfeldt

**Affiliations:** 1Department of Chemistry, Haverford College, Haverford, PA 19041, USA; bpogostin@gmail.com; 2Biochemistry and Structural Biology, Department of Chemistry, Lund University, P.O. Box 124, SE-221 00 Lund, Sweden; malmendal@gmail.com

**Keywords:** Raman spectroscopy, ^1^H NMR spectroscopy, vibrational probes, histidine, peptides, proteins, isotopic labeling, deuterium replacement, acid dissociation constant

## Abstract

Determining the pKa of key functional groups is critical to understanding the pH-dependent behavior of biological proteins and peptide-based biomaterials. Traditionally, ^1^H NMR spectroscopy has been used to determine the pKa of amino acids; however, for larger molecules and aggregating systems, this method can be practically impossible. Previous studies concluded that the C-D stretches in Raman are a useful alternative for determining the pKa of histidine residues. In this study, we report on the Raman application of the C2-D probe on histidine’s imidazole side chain to determining the pKa of histidine in a short peptide sequence. The pKa of the tripeptide was found via difference Raman spectroscopy to be 6.82, and this value was independently confirmed via ^1^H NMR spectroscopy on the same peptide. The C2-D probe was also compared to other Raman reporters of the protonation state of histidine and was determined to be more sensitive and reliable than other protonation-dependent signals. The C2-D Raman probe expands the tool box available to chemists interested in directly interrogating the pKa’s of histidine-containing peptide and protein systems.

## 1. Introduction

There is currently a great deal of published data from infrared (IR) probes [1,2,3]; however, Raman scattering spectroscopy is also gaining traction as a technique for collecting signals from vibrational probe groups. In comparison to IR absorption, complementary selection rules of Raman scattering mean that probe signals can sometimes be more easily observed in Raman than Fourier transform infrared (FTIR) spectroscopy [4]. Additionally, Raman scattering is less sensitive to sample preparation than IR absorption, which often requires experiments to be conducted in D_2_O and in an atmosphere devoid of water vapor and carbon dioxide in order to obtain high-quality spectra.

Carbon-deuterium (C-D) stretches can be employed as a vibrational probe in both FTIR and Raman spectroscopy to report on the molecular environment and dynamics of a selectively deuterated species [1,5,6]. One of the first analyses of C-D bands was in the FTIR and Raman spectra of CD_3_OH and CD_3_OD [7], where exchanging a protium for a deuterium (C-H to C-D) caused a shift of ~800 cm^−1^ for the anti-symmetric and symmetric stretches from 2850–3000 cm^−1^ to 2050–2350 cm^−1^, respectively. The C-D stretches of deuterated methanol also reports on whether the methanol oxygen atom is protonated or deuterated. Other early Raman studies of the C-D stretches investigated the spectra of a series of deuterated glycine derivatives (ND_3_CD_2_COO^−^, NH_3_CD_2_COO^−^, and ND_3_CH_2_COO^−^), and 9 deuterated derivatives of stearic acid [8,9]. The anti-symmetric stretch around 2300 cm^−1^ is significantly more intense in the Raman spectrum compared to in the FTIR spectrum.

Before regular use of the C-D stretch as a vibrational probe in Raman spectroscopy, C-H stretches (ca. 2800–3300 cm^−1^) were employed to investigate the molecular environment and dynamics of lipid membranes [10,11,12,13,14] as well as to study saturated hydrocarbons [15,16]. The C-H stretching region, however, is often obscured by other vibrational modes common in organic compounds, including N-H and O-H stretches (3000–4000 cm^−1^) [8]. C-D stretches, in contrast, appear in a “quiet” window of the Raman spectrum obscured by few other signals save for C≡C (2100–2260 cm^−1^), C≡N (2230–2350 cm^−1^), and cumulative double bonds (C=C=C, C=C=X: 1900–2300 cm^−1^), which are all possible probes themselves and are very uncommon in natural systems [17,18,19]. The change from hydrogen to deuterium is also a minimal structural perturbation, especially compared to the already low perturbativity of other vibration probe groups. Thus, the site-specific incorporation of a C-D probe allows for the isolation and investigation of the signal of a specific C-H bond without significantly altering the structure of the compound of interest.

The C-D probe has been used to report on a variety of changes in structure, dynamics, and chemical environment. IR spectra of selectively deuterated amino acids have been used as probes of folding and dynamics in proteins [1,5,6]. Raman spectra of deuterated methane were used to interrogate organic clathrate hydrate structures that trap methane inside their lattice [20]. Raman C-D stretches have also been used as imaging probes to follow single eukaryotic cell metabolism with high temporal and spatial resolution [21]; by introducing a fully deuterated substrate of interest, such as an amino acid or D_2_O, to the cell media, it is possible to probe its uptake and use by observing the Raman C-D stretching region in live cells [22,23,24].

Calculations by Miller and Corcelli suggested that some C-D frequencies from varying deuterated amino acid side chains could be strongly sensitive to electrostatic changes like protonation and deprotonation events. This idea was consistent with the large experimental response observed classically for deuterated methanol [7] and also suggested that particular C-D stretches might be excellent protonation state sensors (i.e., for histidine residues in proteins) [25].

In this study, we report on the Raman application of the C-D probe to determining the pKa of the amino acid residue histidine in a short peptide sequence. There is much recent interest in pH-responsive synthetic and natural polymers for applications in biomedicine, such as drug delivery, tissue engineering, and biosensors [26,27,28,29], and many cellular proteins behave in a pH-sensitive manner. The ability to determine the pKa of key functional groups in these processes is especially critical to understanding the pH-dependent behavior of peptides [30]. In this context, histidine is one of the most commonly studied amino acid residues. Due to the near neutral pKa of one of the N-bonded protons on the imidazole side chain of histidine (Figure 1), the histidine residue can act both as a base and, when protonated, as an acid, under physiological conditions. The pKa of histidine is sensitive to its specific chemical environment; for example, it has been found that the pKa values of the histidine residues in the protein phosphatidylinositol-specific phospholipase C vary between 5.4-7.6 as determined by ^1^H NMR spectroscopy [31]. Thus, to understand pH-responsive peptides and peptide materials, it is necessary to resolve the pKa of specific histidine residues within these systems.

The deprotonated N atom in histidine’s neutral form is usually the δ-nitrogen on the imidazole side chain (Figure 1): histidine readily exchanges between the Nδ-H and Nε-H tautomers, however the Nε-H (shown in Figure 1) tautomer is favored ~4:1 at high pH when not incorporated in an amino acid sequence. Although this ratio is environmentally dependent, for the purposes of this paper we depict this residue in the form of the Nε-H tautomer [32]. The C2-D stretching frequency cannot distinguish between the two singly-protonated tautomers [30].

Traditionally, titration experiments involving ^1^H NMR spectroscopy have been used to determine the pKa value of amino acids, including that of histidine [31,33,34,35,36,37]; however, alternative methods have also been employed, including capillary zone electrophoresis [38,39] and FTIR using C=O, COO-, and N-O stretches of other titratable functional groups [40]. The ^1^H chemical shifts of the hydrogen atoms attached to C2 and C4 (Figure 1) on the imidazole ring are sensitive to the protonation state of the neighboring δ-nitrogen. Upon deprotonation, the C2-H and C4-H proton signals shift from 8.6 to 7.8 ppm and from 7.4 to 7.0 ppm, respectively [36]. The pKa value of the histidine residue is extracted by plotting the chemical shift as a function of pH and fitting the data to a sigmoidal curve. ^1^H NMR spectroscopy, however, is limited to soluble peptides and proteins with relatively small molecular masses compared to many natural and artificial biomolecules. For larger molecules and aggregating systems, NMR signals are lost due to line broadening as a consequence of structural heterogeneity and/or slow tumbling on the NMR time scale. While solid-state NMR poses a possible alternative option for recovering signals from such samples, the resolution from such measurements largely excludes determination of protonation states.

Raman spectroscopy offers a promising alternative due to its relative insensitivity to sample preparation and its ability to resolve signals from arbitrary phases. The imidazole C4-C5 and C4-N3 stretches, and potentially the C-H stretches, can report on the protonation state of the sidechain of histidine [41,42]. Although these indicators are useful for small molecules like methylimidazole, it is likely that the Raman spectra of larger and more complex peptide sequences would be too crowded to provide the pKa of a single residue with any great clarity. In 2013, Hoffman et al. published an alternative method to determine the pKa of histidine using the C2-D stretch in Raman spectroscopy [30]. The C2-D Raman stretch is clearly sensitive to the protonation state of the N atoms on imidazole: in the doubly protonated state it appears at 2390 cm^−1^, and then shifts 40 cm^−1^ to 2350 cm^−1^ upon deprotonation (Figure 2). This shift was monitored as a function of pH and used to report the pKas of L-carnosine, methylimidazole, and histidine individually in solution. The goal of this study was to further develop and optimize this method and assess if the pKa of histidine could be determined in a more complex system, such as the end-capped peptide Ac-NH-histidine-valine-aspartic acid-CONH_2_ (abbreviated HVD), which is the beginning of a pH-dependent hydrogel-forming sequence [26].

## 2. Results

### 2.1. HVD Deuterium Exchange Reaction

Deuterium incorporation at C2 of the imidazole ring of HVD was accomplished through a previously reported exchange reaction (Figure 3) [30]. The peptide was dissolved in D_2_O to ~10 mM concentration, corrected to pH 8 with NaOD, heated at 50 °C for three days, then lyophilized. Readily exchangeable protons, such as the amide backbone N-H and the O-H of the carboxylic acid sidechain of the aspartic acid, were then back-exchanged via the addition of 0.01 M HCl, followed by re-lyophilization. Mass spectrometry analysis of the post-exchange tripeptide confirmed quantitative exchange of hydrogen to deuterium without further persistent deuteration at other positions, as verified by the increase of the peptide mass by 1 Dalton. Raman spectroscopy further confirmed the back exchange of hydrogen for deuterium at all readily exchangeable positions via the absence of broad N-D and O-D stretches around 2400 cm^−1^ that could overlap with the sharper C2-D stretch in the aqueous Raman spectrum.

The deuterium probe in H(C2-D)VD was stable over the biologically relevant pH of 4–10, and similar in stability to C2-deuterated histidine, as previously reported [30]. No back exchange of C2-D to C2-H was observed at neutral pH when stored at 5–21 °C for several months.

### 2.2. pKa Determination

To explore the efficacy of the method of Hoffman et al. in larger peptide sequences, a 10 mM solution in H_2_O of C2-deuterated HVD was titrated with NaOD to achieve increments of 0.5 or 0.25 pH units, from pH 4.5 to pH 9, to determine the pKa of the histidine residue. In the case of the tripeptide, the individual peaks at 2360 cm^−1^ and 2390 cm^−1^ were assigned to the neutral and protonated forms of histidine, respectively (Figure 4). It is interesting to note that the C2-D peak corresponding to the deprotonated form of histidine in HVD was shifted by 10 cm^−1^ from the value previously reported for histidine alone (2350 cm^−1^) [30]. This shift is perhaps due to differential electrostatic effects associated with the nearby presences of the protonated amino acid terminus and negatively charged carboxyl group in the free amino acid compared to amide bonds in those same positions for the capped peptide.

Although the C2-D signals were present, they were too weak in the Raman spectra to be analyzed directly in the manner described by Hoffman et al. The pKa of HVD was instead determined by creating Raman difference spectra (Appendix A) for each pH measurement (using the pH 4 spectrum as reference) which were then superimposed (Figure 4). The difference spectra highlight the contrast between the protonated and deprotonated C2-D signals and thereby allow the data to be more readily analyzed. The maximum intensity of the low frequency peak at 2362 cm^−1^ (where the neutral species grows in) for each difference spectrum was then plotted against pH, and the data were then fitted to a sigmoidal curve (Appendix A). The low frequency peak is higher in intensity than the high frequency peak: this increase in intensity makes this signal more sensitive and easier to analyze (as reported previously in [30]). The midpoint of the resulting sigmoidal curve, which corresponds to the pKa of the histidine residue in the tripeptide, was 6.82 (see Figure 5).

This pKa is slightly higher than that of histidine in solution (ca. 6) [43]. The pKa of histidine can shift depending on the chemical environment of the residue [31]; the difference in pKa value here is thus likely due to the unique chemical environment of this residue in HVD, where among other factors, the histidine residue is in relatively close proximity to aspartic acid. The aspartic acid carboxylate side chain at neutral pH could stabilize the protonated form of histidine and thus increase its pKa value.

The pKa determined by Raman difference spectroscopy was confirmed by a ^1^H NMR titration experiment on non-deuterated peptide under the same conditions. By following the chemical shifts of the hydrogens attached to C2 and C4 of the imidazole ring [36] of the tripeptide, the pKa of histidine in HVD was found to be 6.87 (Figure 6), which is within 0.05 pH units of and statistically identical (*p* = 0.95 assuming Gaussian error) to the pKa determined by Raman spectroscopy. These ^1^H NMR results confirm that Raman spectroscopy is an equally accurate method to determine the pKa of histidine in the peptide.

## 3. Discussion

This study demonstrates clearly that Raman spectroscopy provides a reliable method to determine the pKa of histidine. A method first developed by Hoffman et al. for the determination of the pKa of imidazole is expanded here to demonstrate its utility in peptides. This method provides a particularly compelling means for determining the pKa of histidine in systems where ^1^H NMR spectroscopy titrations are of limited use, including peptide-based materials that undergo aggregation and/or phase transitions in response to pH changes (like the parent sequence of the HVD peptide). Such systems include, but are not limited to, various amyloid-forming proteins, such as α-synuclein [44,45,46,47], and peptide-based materials, such as hydrogels, which have recently been popularized due to their potential medical applications as tissue scaffolds and drug delivery systems [26,29,48]. Hydrogels and other soft-matter peptide materials exist at the solid-liquid interface, and as such, Raman spectroscopy offers a flexible method to probe the pKa of histidine in all states of these materials. For most of these examples, ^1^H NMR spectroscopy is inadequate due to sample constraints, as previously discussed.

Previous studies noted that other modes of the imidazole ring, particularly C4-C5 (~1600 cm^−1^) and C4-N3 (~1100 cm^−1^) stretching, can report on the protonation state of histidine [41]. A possible benefit of the C4-C5 and C4-N3 stretches over C2-D is that they are more intense in the Raman spectrum, and thus the difference spectra used in Figure 5 might not need to be constructed to view these signals. Those regions in the Raman spectrum of HVD were also explored to evaluate if those signals might report on pKa of histidine in this relatively simple peptide.

A peak at 1115 cm^−1^ in the Raman spectrum was tentatively assigned to the C4-N3 stretch, but as shown in Appendix A, for this peptide and in contrast to methylimidazole, this signal does not consistently track with changes in pH. The C4-C5 stretch observed at 1580 cm^−1^ exhibits a more consistent pH-response (Figure 7); however, the data in Figure 7 did not provide as good a fit to a sigmoidal curve (Appendix A). The histidine C4-C5 stretch for HVD is expected to overlap with other C-C stretches within the sequence and/or the peptide’s amide II backbone signals [42,49]. Therefore, while there is a relatively clear pH-response from the C4-C5 stretch that appears to report on the histidine protonation state, there may be other signals that also respond to pH that complicate the interpretability of this spectral region. The C2-D signal, in spite of its weaker intensity, proved to be a more sensitive and direct reporter compared to other possible Raman probes in HVD and is also likely the best candidate for determining the pKa of histidine in longer peptides or proteins with even more congested vibrational spectra.

While the C2-D Raman probe expands the tool box available to those interested in directly interrogating the pKa of histidine-containing peptide and protein systems, its general utility will depend to a large extent on its relatively weak signal. We note that in these experiments we are still using a relatively unsophisticated, non-resonant CW-Raman instrument; stimulated Raman or other pulsed-laser techniques with much higher peak powers might expand the applicability of this very specific probe group to lower peptide or protein concentrations and more complicated samples. HVD is the beginning of a 13-amino acid sequence that forms hydrogels, and that sequence is the next target (in both liquid and gelled forms) for this probe technique.

## 4. Materials and Methods

### 4.1. Materials

All solvents and reagents were purchased from commercial suppliers and used without further purification. 

Electrospray mass spectrometry was performed at Haverford College on an Agilent (Santa Clara, CA, USA) 1100 LC/ES Mass Spectrometervia direct infusion of diluted low-pH solutions into the electrospray chamber. See [App app2dot1-molecules-24-00405].

### 4.2. Peptide Synthesis and Purification

The tripeptide CH_3_CONH-HVD-CONH_2_ was made by solid phase peptide synthesis using standard Fmoc (*N*-(9-fluorenyl)methoxycarbonyl) chemistry protocol employing an Applied Biosystems 433A synthesizer (Company, City, State abbreviation if CA or USA, Country). RINK resin (0.1 mmol) (Company, City, State abbreviation if CA or USA, Country) was used to provide a C-terminal carboxamide.

The crude tripeptide was purified by HPLC using a reversed phase C_8_ preparative column. The identity and purity of the resulting peptide was verified by electrospray mass spectrometry (ES MS) with the major peak at *m/z* [M + H] = 411 amu (Appendix A, top). See [App app2dot1-molecules-24-00405].

### 4.3. Deuterium Exchange Reaction

The tripeptide, CH_3_CONH-HVD-CONH_2_ (8.80 mg; 0.0214 mmol) was dissolved in 1.5 mL of deuterium oxide, D_2_O, (99.8%) in a 2 mL Eppendorf tube and raised to pH 8 with sodium deuterium oxide (NaOD, 40%) and deuterium chloride (DCl, 35%) and measured with a pH electrode. The solution was then placed in a 50 °C water bath for ~72 h, frozen with dry ice, and lyophilized overnight. The resulting white powder was dissolved in 1.5 mL 0.01 M hydrochloric acid (HCl) to remove all readily exchangeable residual deuterons, then frozen and lyophilized. The selective incorporation of a single deuterium atom was confirmed by ES MS (*m/z* 412) with disappearance of the peak at *m/z* 411, confirming complete exchange (Appendix A).

### 4.4. Raman Sample Preparation

Deuterated tripeptide HVD (8.80 mg; 0.0214 mmol) was dissolved in 1.5 mL water and split into three 0.5 mL peptide stock solutions. Two of these stock solutions were used for the titration experiment, as explained in more detail below, while the third, of the exact same identity as the other two, was refrigerated as a backup. The titration of the tripeptide occurred over the pH range of 4.0–9.0, and the pH was increased by 0.5 pH units, except between pH 6.50–8.50 where the step size was decreased to 0.25 to provide more data near the expected pKa value. The stock solution was corrected to the desired pH by adding 0.2–2 μL aliquots of NaOH or HCl (0.1–5 M), as needed. After the desired pH was achieved, a sample for Raman spectroscopy was prepared by placing approximately 5 μL of solution into a 1 mm glass capillary tube. This process of correcting the pH of the stock solution and preparing the Raman sample was repeated for the duration of the titration. To prevent the concentration of the samples from changing by more than 5%, one peptide stock solution was used for the first half of the titration experiment and a second stock solution was used for the remaining pH points.

### 4.5. Raman Spectroscopy

Raman scattering experiments were performed with a home-built continuous wave Raman spectrometer reported previously [30] but with modification since that initial report. The excitation source here was a Cobolt, Inc. DPSS 532 nm laser attenuated to incident power of 200 mW. Samples were loaded into 1 mm glass capillary tubes (held at a constant temperature of 22 °C using a home-built circulating sample holder) and excited vertically across the width of the tubes. Scattered light was collected at 90° to the incident excitation beam via a Nikon camera lens, focused into a SpectraPro 0.5 m monochromator (using a 600 grooves/mm grating blazed at 500 nm), and collected on a PI-Acton Spec10/100 liquid N_2_-cooled CCD camera (Appendix A). Data was collected with the camera centered at 603 nm over a period of 12 h in accumulations of 1 min. No obvious sample degradation was observed under these conditions, as monitored via comparisons of other regions of the Raman spectrum under lower laser power and shorter exposure times.

### 4.6. Raman Data Processing

All spectra were processed by first subtracting a H_2_O background spectrum normalized to match the intensity of the OH stretching band. Difference spectra were achieved by subtracting the pH 4 HVD standard spectrum from every subsequent spectrum collected (Appendix A). The pH-invariant C-H Raman stretch at 2950 cm^−1^ was used to normalize peak intensities for the peptide in both spectra. The baseline of the difference spectrum was then smoothed by fitting the data from 2270 cm^−1^ to 2420 cm^−1^ to a 3^rd^ degree polynomial and subtracting this fitted line from the data to achieve a final difference spectrum, which results in some sections of the Raman spectrum appearing to have negative intensities/partially dispersive lineshapes due to the polynomial nature of the baseline fit near the actual C2-D stretching signals.

The difference spectra were then overlaid, and the maximum peak intensity at 2362 cm^−1^ of each difference spectrum was plotted versus pH. This data was then fit to a sigmoidal dose-response curve (Equation (1)). The pKa was extracted from the data by finding the midpoint $\left(Log\;x_{0}\right)$ of this sigmoidal curve from its equation. (The *p* parameter here is the Hill-type “slope” or “cooperativity” parameter and does not influence the midpoint, which is different from the “*p*” probability measure mentioned above when comparing experimentally determined midpoints.)
(1)$$y=A_{1}+\frac{\left(A_{2}-A_{1}\right)}{1+10^{(\log\left(x_{0})-x\right)*p}}$$

### 4.7. ^1^H NMR Spectroscopy

^1^H NMR spectroscopy samples were 1 mM of HVD in ddH_2_O doped with 10% D_2_O and 0.1% 4,4-dimethyl-4-silapentane-1-sulfonic acid (DSS) as a chemical shift standard. Experiments were performed in a quartz tube at 25 °C using an Agilent VNMRS DirectDrive (Agilent, Santa Clara, CA, USA) spectrometer at Lund University, Sweden operating at a ^1^H frequency of 600 MHz. The chemical shifts of the histidine C2 and C4 hydrogens were monitored via 13 1D ^1^H spectra acquired over the pH range from 3.2 to 9.0. Spectra were recorded with 16 scans and a total acquisition time of 1 min. Residual water was suppressed by steady state pulses. Spectra were processed with iNMR (Nucleomatica, Molfetta, Italy).

## Figures and Tables

**Figure 1 molecules-24-00405-f001:**
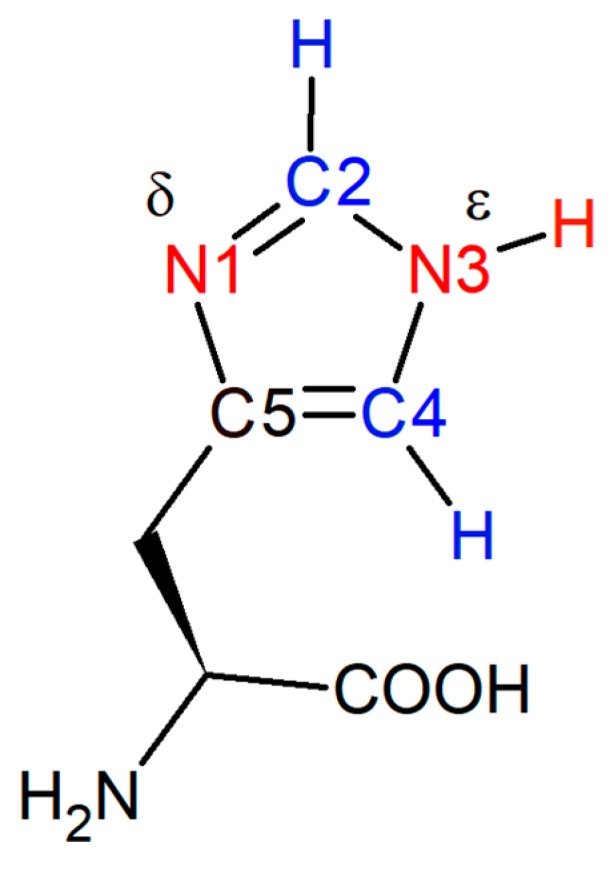
Histidine with carbon atoms 2 and 4 (blue) and δ- and ε-nitrogen atoms (red) on the imidazole ring. The pKa of the protonated δ-nitrogen is near neutral, and the chemical shifts of protons attached to C2 and C4 in ^1^H NMR spectroscopy are frequently used to determine its pKa.

**Figure 2 molecules-24-00405-f002:**
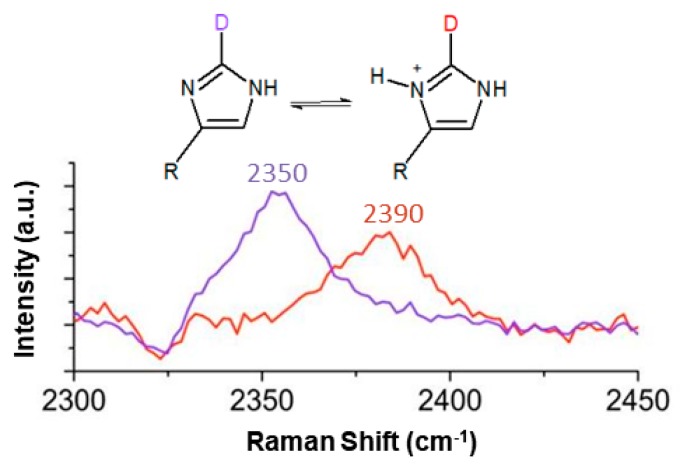
The C2-D Raman stretch from the imidazole sidechain of histidine is sensitive to the protonation state of the neighboring N atoms. The protonated and deprotonated imidazole ring C2-D stretches were observed at 2390 cm^−1^ (red curve; pH 2.5) and 2350 cm^−1^ (purple curve; pH 9.8), respectively. Adapted with permission from [30]. Copyright 2013 American Chemical Society.

**Figure 3 molecules-24-00405-f003:**
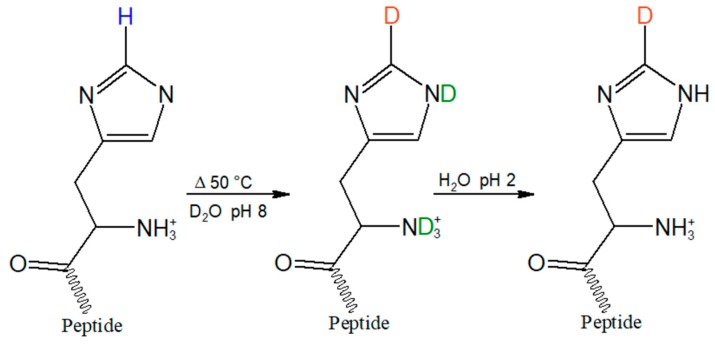
Incorporation of the C2-D probe (orange) on the histidine imidazole ring was achieved by heating the tripeptide in D_2_O for three days at 50 °C in D_2_O at pH 8. To back exchange readily exchangeable hydrogens on nitrogen and oxygen (green), the sample was then re-dissolved in H_2_O, pH 2, and lyophilized.

**Figure 4 molecules-24-00405-f004:**
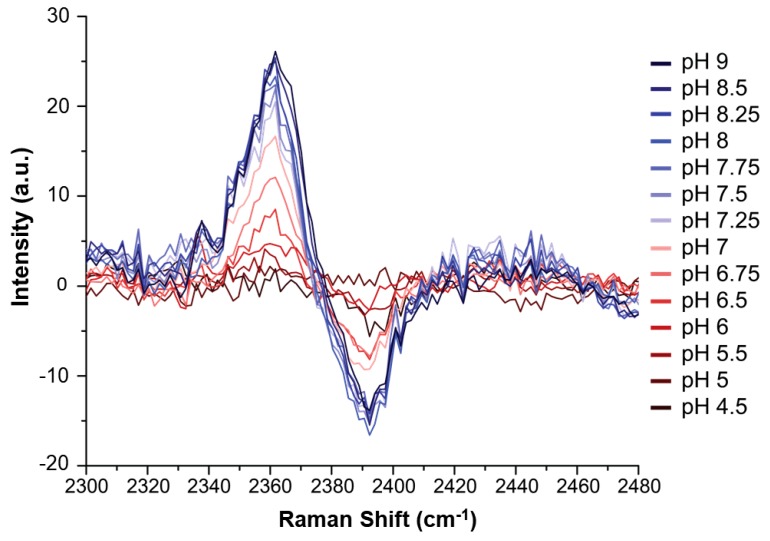
Raman difference spectra of H(C2-D)VD peptide at each pH value, versus the spectrum at pH 4. More basic samples (from red to blue) display stronger difference spectra up through pH 8.25 (thus indicating the end of the titration).

**Figure 5 molecules-24-00405-f005:**
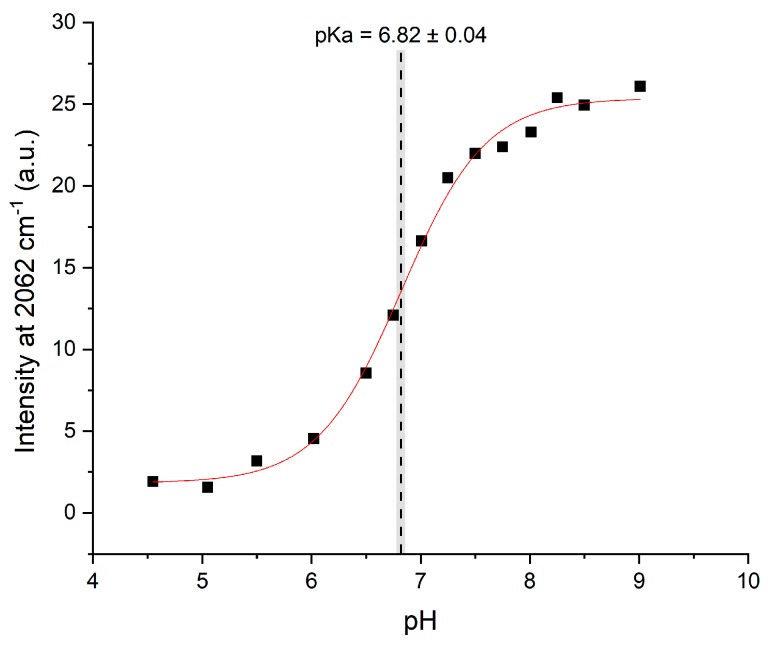
The intensity of the difference spectra at 2362 cm^−1^ for each titration point vs pH, fitted to a dose-response sigmoidal curve (Equation S1). The gray bar represents the error in the fitted midpoint.

**Figure 6 molecules-24-00405-f006:**
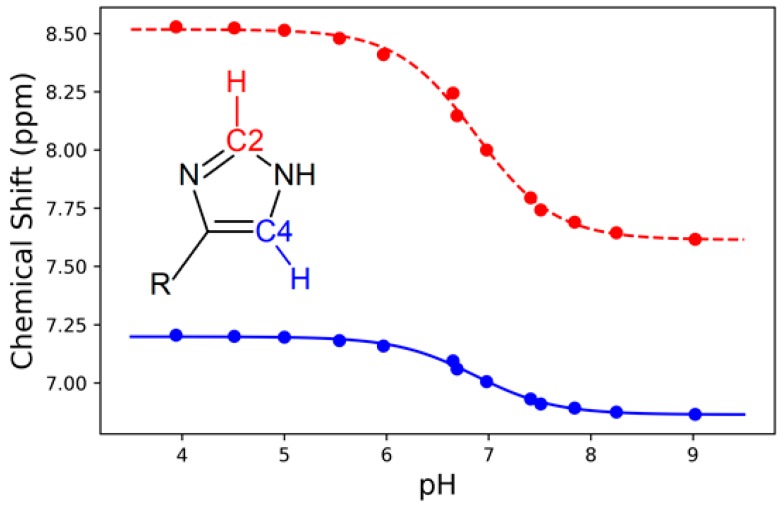
The ^1^H chemical shift for the hydrogen atoms attached to C2 (red) and C4 (blue) of the imidazole ring versus pH, fitted to a dose-response sigmoidal curve (Equation (1)). The average midpoint for the C2 and C4 sigmoidal curves yielded a pKa of 6.87 ± 0.03.

**Figure 7 molecules-24-00405-f007:**
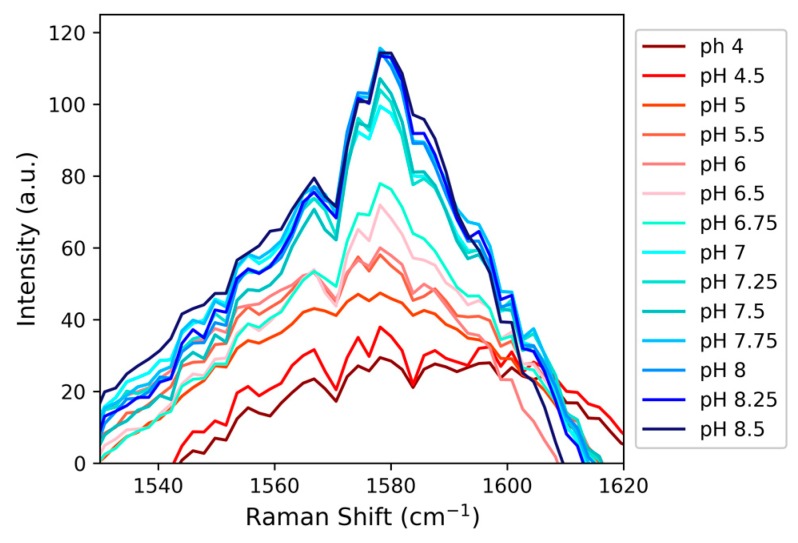
The growth of a peak at 1580 cm^−1^, circumstantially assigned to the C4-C5 stretch from imidazole (based on prior precedent), as a function of pH demonstrates that the signal is sensitive to changes in the protonation state of histidine.

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
