# Peer review of "pKa Determination of a Histidine Residue in a Short Peptide Using Raman Spectroscopy"

_molecules, 2019, doi:10.3390/molecules24030405_

Round 1

Reviewer 1 Report

The MS reports on the study of His pH dependence developing a new indicator of the protic equilibrium using C2-D stretching mode assessible via Raman spectroscopy. The authors claim that NMR spectroscopy would have difficulties in evaluating electrostatic environment of a specific His residue in longer peptides, while Raman spectroscopy would focus specifically on the labeled His. While there is a concern that Raman spectra presented are noisy and require long time to accumulate (compared to NMR), but the ability to gain required information overweighs the complexity. The results are solid and trustable; the paper is well written. I recommend the MS for publication to Molecules, given that the comments are addressed.

The authors fit the data in Fig. 5 to equation 1. In addition to pKa, there is additional parameter worth discussing; that is p.  The authors should give its value and discuss its meaning.

The authors claim that the two tautomers cannot be distinguished by C2D stretching frequency but provided no explanation nor a reference.

The authors attempted using CC stretching mode for a similar purpose but rejected rather strongly this label saying that the pKa obtained doesn’t match other measurements. There are two reasons why the resulting pKa may be different, one is the quality of the measurements (mentioned by the authors), another is the local environment. I would like to see the assessment of the reason for the observation. Also, what is the reason the high pH values were not assessed in these measurements?

No information on the sample degradation in Raman measurements is given.

Figure 2 caption:  pH conditions for the two spectra are required.

It is not clear why the authors used only the low-frequency peak (Fig. 4) in assessing the equilibrium.  

Supplementary Equations S1 has a negative sign for the second term, which should be positive for Fig. 5. The precision for the numbers in S1 should not go wild (currently 10 significant figures).

It is surprising to see negative Raman intensities (Fig. 7 and Fig. S3). Please explain and/or correct.

Reviewer 2 Report

The manuscript entitled “pKa Determination of a Histidine Residue in a Short Peptide Using Raman Spectroscopy” describes the determination of Pka value of histidine in short peptide using Raman spectroscopy rather than describes its NMR studies, which is a good hypothesis. Demonstrating about the pKa value histidine, using IR spectral C-N, C-C and C-D stretching techniques of histidine and comparison study with Raman spectroscopy and observation high sensitivity of Raman spectra about these stretching’s to determine the pKa value of histidine in a short peptide is an awesome presentation. Even though pKa values is determined by NMR spectroscopy for histidine, Raman spectroscopy is still more useful and sensitive technique. It is an interesting piece of study to determine the pKa value of histidine I a short peptide.
